# Opportunistic Learning:
# Budgeted Cost-Sensitive Learning from Data Streams

**Mohammad Kachuee, Orpaz Goldstein, Kimmo Karkkainen, Sajad Darabi, Majid Sarrafzadeh**
Department of Computer Science
University of California, Los Angeles (UCLA)
Los Angeles, CA 90095, USA
{mkachuee,orpgol,kimmo,sajad.darabi,majid}@cs.ucla.edu

## Abstract

In many real-world learning scenarios, features are only acquirable at a cost constrained under a budget. In this paper, we propose a novel approach for cost-sensitive feature acquisition at the prediction-time. The suggested method acquires features incrementally based on a context-aware feature-value function. We formulate the problem in the reinforcement learning paradigm, and introduce a reward function based on the utility of each feature. Specifically, MC dropout sampling is used to measure expected variations of the model uncertainty which is used as a feature-value function. Furthermore, we suggest sharing representations between the class predictor and value function estimator networks. The suggested approach is completely online and is readily applicable to stream learning setups. The solution is evaluated on three different datasets including the well-known MNIST dataset as a benchmark as well as two cost-sensitive datasets: Yahoo Learning to Rank and a dataset in the medical domain for diabetes classification. According to the results, the proposed method is able to efficiently acquire features and make accurate predictions.

## 1 Introduction

In traditional machine learning settings, it is usually assumed that a training dataset is freely available and the objective is to train models that generalize well. In this paradigm, the feature set is fixed, and we are dealing with complete feature vectors accompanied by class labels that are provided for training. However, in many real-world scenarios, there are certain costs for acquiring features as well as budgets limiting the total expenditure. Here, the notation of cost is more general than financial cost and it also refers to other concepts such as computational cost, privacy impacts, energy consumption, patient discomfort in medical tests, and so forth (Krishnapuram et al., 2011). Take the example of the disease diagnosis based on medical tests. Creating a complete feature vector from all the relevant information is synonymous with conducting many tests such as MRI scan, blood test, etc. which would not be practical. On the other hand, a physician approaches the problem by asking a set of basic easy-to-acquire features, and then incrementally prescribes other tests based on the current known information (i.e., context) until a reliable diagnosis can be made. Furthermore, in many real-world use-cases, due to the volume of data or necessity of prompt decisions, learning and prediction should take place in an online and stream-based fashion. In the medical diagnosis example, it is consistent with the fact that the latency of diagnosis is vital (e.g., urgency of specific cases and diagnosis), and it is often impossible to defer the decisions. Here, by online we mean processing samples one at a time as they are being received.

Various approaches were suggested in the literature for cost-sensitive feature acquisition. To begin with, traditional feature selection methods suggested to limit the set of features being used for training (Greiner et al., 2002; Ji & Carin, 2007). For instance, L1 regularization for linear classifiers results in

---

A version of the source code and the health dataset preproccessing code for this paper is available at:
https://github.com/mkachuee/Opportunistic

models that effectively use a subset of features (Efron et al., 2004). Note that these methods focus on finding a fixed subset of features to be used (i.e., feature selection), while a more optimal solution would be making feature acquisition decisions based on the sample at hand and at the prediction-time.

More recently, probabilistic methods were suggested that measure the value of each feature based on the current evidence (Chen et al., 2015). However, these methods are usually applicable to Bayesian networks or similar probabilistic models and make limiting assumptions such as having binary features and binary classes (Chen et al., 2014). Furthermore, these probabilistic methods are computationally expensive and intractable in large scale problems (Chen et al., 2015).

Motivated by the success of discriminative learning, cascade and tree based classifiers suggested as an intuitive way to incorporate feature costs (Karayev et al., 2012; Chen et al., 2012; Xu et al., 2012; 2014). Nevertheless, these methods are basically limited to the modeling capability of tree classifiers and are limited to fixed predetermined structures. A recent work by Nan & Saligrama (2017) suggested a gating method that employs adaptive linear or tree-based classifiers, alternating between low-cost models for easy-to-handle instances and higher-cost models to handle more complicated cases. While this method outperforms many of the previous work on the tree-based and cascade cost-sensitive classifiers, the low-cost model being used is limited to simple linear classifiers or pruned random forests.

As an alternative approach, sensitivity analysis of trained predictors is suggested to measure the importance of each feature given a context (Early et al., 2016a; Kachuee et al., 2017; 2018). These approaches either require an exhaustive measurement of sensitivities or rely on approximations of sensitivity. These methods are easy to use as they work without any significant modification to the predictor models being trained. However, theoretically, finding the global sensitivity is a difficult and computationally expensive problem. Therefore, frequently, approximate or local sensitivities are being used in these methods which may cause not optimal solutions.

Another approach that is suggested in the literature is modeling the feature acquisition problem as a learning problem in the imitation learning (He et al., 2012) or reinforcement learning (He et al., 2016; Shim et al., 2017; Janisch et al., 2017) domain. These approaches are promising in terms of performance and scalability. However, the value functions used in these methods are usually not intuitive and require tuning hyper-parameters to balance the cost vs. accuracy trade-off. More specifically, they often rely on one or more hyper-parameters to adjust the average cost at which these models operate. On the other hand, in many real-world scenarios it is desirable to adjust the trade-off at the prediction-time rather than the training-time. For instance, it might be desirable to spend more for a certain instance or continue the feature acquisition until a desired level of prediction confidence is achieved.

This paper presents a novel method based on deep Q-networks for cost-sensitive feature acquisition. The proposed solution employs uncertainty analysis in neural network classifiers as a measure for finding the value of each feature given a context. Specifically, we use variations in the certainty of predictions as a reward function to measure the value per unit of the cost given the current context. In contrast to the recent feature acquisition methods that use reinforcement learning ideas (He et al., 2016; Shim et al., 2017; Janisch et al., 2017), the suggested reward function does not require any hyper-parameter tuning to balance cost versus performance trade-off. Here, features are acquired incrementally, while maintaining a certain budget or a stopping criterion. Moreover, in contrast to many other work in the literature that assume an initial complete dataset (He et al., 2012; Kusner et al., 2014; Chen et al., 2015; Early et al., 2016b; Nan & Saligrama, 2017), the proposed solution is stream-based and online which learns and optimizes acquisition costs during the training and the prediction. This might be beneficial as, in many real-world use cases, it might be prohibitively expensive to collect all features for all training data. Furthermore, this paper suggests a method for sharing the representations between the class predictor and action-value models that increases the training efficiency.

## 2 PRELIMINARIES

### 2.1 PROBLEM SETTINGS

In this paper, we consider the general scenario of having a stream of samples as input ($S_i$). Each sample $S_i$ corresponds to a data point of a certain class in $\mathbb{R}^d$, where there is a cost for acquiring each feature ($c_j; 1 \leq j \leq d$). For each sample, initially, we do not know the value of any feature. Subsequently, at each time step $t$, we only have access to a partial realization of the feature vector denoted by $\boldsymbol{x}_i^t$ that consists of features that are acquired so far. There is a maximum feature acquisition budget ($B$) that is available for each sample. Note that the acquisition may also be terminated before reaching the maximum budget, based on any other termination condition such as reaching a certain prediction confidence. Furthermore, for each $S_i$, there is a ground truth target label $\tilde{y}_i$. It is also worth noting that we consider the online stream processing task in which acquiring features is only possible for the current sample being processed. In other words, any decision should take place in an online fashion.

In this setting, the goal of an Opportunistic Learning (OL) solution is to make accurate predictions for each sample by acquiring as many features as necessary. At the same time, learning should take place by updating the model while maintaining the budgets. Please note that, in this setup, we are assuming that the feature acquisition algorithm is processing a stream of input samples and there are no distinct training or test samples. However, we assume that ground truth labels are only available to us after the prediction and for a subset of samples.

More formally, we define a mask vector $\boldsymbol{k}_i^t \in \{0,1\}^d$ where each element of $\boldsymbol{k}$ indicates if the corresponding feature is available in $\boldsymbol{x}_i^t$. Using this notation, the total feature acquisition cost at each time step can be represented as

$$C_{total,i}^t = (\boldsymbol{k}_i^t - \boldsymbol{k}_i^0)^T \boldsymbol{c} \ . \tag{1}$$

Furthermore, we define the feature query operator (q) as

$$\boldsymbol{x}_i^{t+1} = q(\boldsymbol{x}_i^t, j), \text{ where } \boldsymbol{k}_{i,j}^{t+1} - \boldsymbol{k}_{i,j}^t = 1 \ . \tag{2}$$

In Section 3, we use these primitive operations and notations for presenting the suggested solution.

### 2.2 PREDICTION CERTAINTY

As prediction certainty is used extensively throughout this paper, we devote this section to certainty measurement. The softmax output layer in neural networks are traditionally used as a measure of prediction certainty. However, interpreting softmax values as probabilities is an ad hoc approach prone to errors and inaccurate certainty estimates (Szegedy et al., 2013). In order to mitigate this issue, we follow the idea of Bayesian neural networks and Monte Carlo dropout (MC dropout) (Williams, 1997; Gal & Ghahramani, 2016). Here we consider the distribution of model parameters at each layer $l$ in an $L$ layer neural network as:

$$\hat{\omega}_l \sim p(\omega_l), \text{ where } 1 \leq l \leq L \ , \tag{3}$$

where $\hat{\omega}_l$ is a realization of layer parameters from the probability distribution of $p(\omega_l)$. In this setting, a probability estimate conditioned on the input and stochastic model parameters is represented as:

$$p(y|\boldsymbol{x}, \hat{\omega}) = \text{softmax}(f_{\mathbb{D}}^{\hat{\omega}}(\boldsymbol{x})) \ , \tag{4}$$

where $f_{\mathbb{D}}^{\hat{\omega}}$ is the output activation of a neural network with parameters $\hat{\omega}$ trained on dataset $\mathbb{D}$. In order to find the uncertainty of final predictions with respect to inputs, we integrate equation 4 with respect to $\omega$:

$$p(y|\boldsymbol{x}, \mathbb{D}) = \int p(y|\boldsymbol{x}, \omega) p(\omega|\mathbb{D}) d\omega \ . \tag{5}$$

Finally, MC dropout suggests interpreting the dropout forward path evaluations as Monte Carlo samples ($\overline{\omega}_t$) from the $\omega$ distribution and approximating the prediction probability as:

$$p(y|\boldsymbol{x}, \mathbb{D}) = \frac{1}{T} \sum_{t=1}^{T} p(y|\boldsymbol{x}, \overline{\omega}_t) \ . \tag{6}$$

With reasonable dropout probability and number of samples, the MC dropout estimate can be considered as an accurate estimate of the prediction uncertainty. Readers are referred to Gal & Ghahramani (2016) for a more detailed discussion. In this paper, we denote the certainty of prediction for a given sample ($Cert(\boldsymbol{x}_i^t)$) as a vector providing the probability of the sample belonging to each class in equation 6.

## 3 PROPOSED SOLUTION

### 3.1 COST-SENSITIVE FEATURE ACQUISITION

We formulate the problem at hand as a generic reinforcement learning problem. Each episode is basically consisting of a sequence of interactions between the suggested algorithm and a single data instance (i.e., sample). At each point, the current state is defined as the current realization of the feature vector (i.e., $\boldsymbol{x}_i^t$) for a given instance. At each state, the set of valid actions consists of acquiring any feature that is not acquired yet (i.e., $A_i^t = \{j = 1 \dots d | \boldsymbol{k}_{i,j}^t = 0\}$). In this setting, each action along with the state transition as well as a reward, defined in the following, is characterizing an experience.

We suggest incremental feature acquisition based on the value per unit cost of each feature. Here, the value of acquiring a feature is defined as the expected amount of change in the prediction uncertainty that acquiring the feature causes. Specifically, we define the value of each unknown feature as:

$$r_{i,j}^t = \frac{||Cert(\boldsymbol{x}_i^t) - Cert(q(\boldsymbol{x}_i^t, j))||}{\boldsymbol{c}_j} \quad , \tag{7}$$

where $r_{i,j}^t$ is the value of acquiring feature $j$ for sample $i$ at time step $t$. It can be interpreted as the expected change of the hypothesis due to acquiring each feature per unit of the cost. Other reinforcement learning based feature acquisition methods in the literature usually use the final prediction accuracy and feature acquisition costs as components of reward function (He et al., 2016; Shim et al., 2017; Janisch et al., 2017). However, the reward function of equation 7 is modeling the weighted changes of hypothesis after acquiring each feature. Consequently, it results in an incremental solution which is selecting the most informative feature to be acquired at each point. As it is demonstrated in our experiments, this property is particularly beneficial when a single model is to be used under a budget determined at the prediction-time or any other, not predefined, termination condition.

While it is possible to directly use the measure introduced in equation 7 to find features to be acquired at each time, it would be computationally expensive; because it requires exhaustively measuring the value function for all features at each time. Instead, in addition to a predictor model, we train an action value (i.e., feature value) function which estimates the gain of acquiring each feature based on the current context. For this purpose, we follow the idea of the deep Q-network (DQN) (Mnih et al., 2015; 2013). Briefly, DQN suggests end-to-end learning of the action-value function. It is achieved by exploring the space through taking actions using an $\epsilon$-greedy policy, storing experiences in a replay memory, and gradually updating the value function used for exploration. Due to space limitations, readers are referred to Mnih et al. (2015) for a more detailed discussion.

Figure 1 presents the network architecture of the proposed method for prediction and feature acquisition. In this architecture, a predictor network (P-Network) is trained jointly with an action value network (Q-Network). The P-Network is responsible for making prediction and consists of dropout layers that are sampled in order to find the prediction uncertainty. The Q-Network estimates the value of each unknown feature being acquired.

Here, we suggest sharing the representations learned from the P-Network with the Q-Network. Specifically, the activations of each layer in the P-Network serve as input to the adjacent layers of the Q-Network (see Figure 1). Note that, in order to increase model stability during the training, we do not allow back-propagation from Q-Network outputs to P-Network weights. We also explored other architectures and sharing methods including using fully-shared layers between P- and Q-Networks that are trained jointly or only sharing the first few layers. According to our experiments, the suggested sharing method of Figure 1 is reasonably efficient, while introducing a minimal burden on the prediction performance.

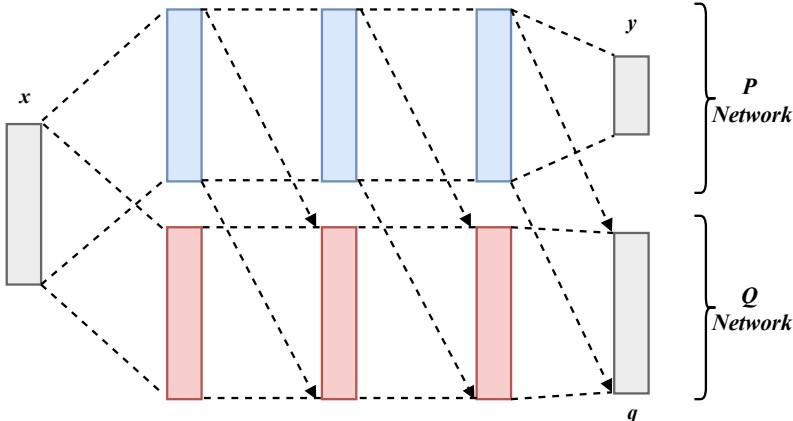

Figure 1: Network architecture of the proposed approach for prediction and action value estimation.

Algorithm 1 summarizes the procedures for cost-sensitive feature acquisition and training the networks. This algorithm is designed to operate on a stream of input instances, actively acquire features, make predictions, and optimize the models. In this algorithm, if any features are available for free we include them in the initial feature vector; otherwise, we start with all features not being available initially. Here, the feature acquisition is terminated when either a maximum budget is exceeded, a user-defined stopping function decides to stop, or there is no unknown feature left to acquire. It is worth noting that, in Algorithm 1, to simplify the presentation, we assumed that ground-truth labels are available at the beginning of each episode. However, in the actual implementation, we store experiences within an episode in a temporary buffer, excluding the label. At last, after the termination of the feature acquisition procedure, a prediction is being made and upon the availability of label for that sample, the temporary experiences along with the ground-truth label are pushed to the experience replay memory.

In our experiments, for the simplicity of presentation, we assume that all features are independently acquirable at a certain cost, while in many scenarios, features are bundled and acquired together (e.g., certain clinical measurements). However, it should be noted that the current formulation presented in this paper allows for having bundled feature sets. In this case, each action would be acquiring each bundle and the reward function is evaluated for the acquisition of the bundle by measuring the variations of uncertainty before and after acquiring the bundle.

### 3.2 Implementation Details

In this paper, PyTorch numerical computational library (Paszke et al., 2017) is used for the implementation of the proposed method. The experiments took between a few hours to a couple days on a GPU server, depending on the experiment. Here, we explored fully connected multi-layer neural network architectures; however, the approach taken can be readily applied to other neural network and deep learning architectures. We normalize features prior to our experiments statistically ($\mu = 0, \sigma = 1$) and impute missing features with zeros. Note that, in our implementation, for efficiency reasons, we use NaN (not a number) values to represent features that are not available and impute them with zeros during the forward/backward computation.

Cross-entropy and mean squared error (MSE) loss functions were used as the objective functions for the P and Q networks, respectively. Furthermore, the Adam optimization algorithm Kingma & Ba (2014) was used throughout this work for training the networks. We used dropout with the probability of $0.5$ for all hidden layers of the P-Network and no dropout for the Q-Network. The target Q-Network was updated softly with the rate of $0.001$. We update P, Q, and target Q networks every $1 + \frac{n_{fe}}{100}$ experiences, where $n_{fe}$ is the total number of features in an experiment. In addition, the replay memory size is set to store $1000 \times n_{fe}$ most recent experiences. The random exploration probability is decayed such that eventually it reaches the probability of $0.1$. We determined these hyper-parameters using the validation set. Based on our experiments, the suggested solution is not much sensitive to these values and any reasonable setting, given enough training iterations, would result in reasonable a performance. A more detailed explanation of implementation details for each specific experiment is provided in Section 4.

---

**Algorithm 1:** Suggested algorithm for Cost-Sensitive Feature Acquisition, Prediction, and Training

---

**Input:** total budget ($B$), stream of samples ($S_i$), acquisition cost of features ($c_j$)
**Initialize:** experience replay memory, random exploration probability ($Pr_{rand}$) $\leftarrow 1$

1   **for** $S_i$ *in the stream* **do**
2     $Pr_{rand} \leftarrow decay\_factor \times Pr_{rand}$
3     $t \leftarrow 0$
4     $\boldsymbol{x}_i^t \leftarrow$ known features of $S_i$        // if there are any features available
5     $total\_cost \leftarrow 0$
6     $\tilde{y}_i \leftarrow$ class label of $S_i$
7     $terminate\_flag \leftarrow False$
8     **while** *not terminate_flag* **do**       // collect experiences from each episode
9        **if** *new random in* $[0, 1) < Pr_{rand}$ **then**    // if it is a random exploration
10           $j \leftarrow$ index of a randomly selected unknown feature
11        **else**                   // if it is an exploration using policy
12           $j \leftarrow$ index of the unknown feature with maximum Q value
13        $\boldsymbol{x}_i^{t+1} \leftarrow q(\boldsymbol{x}_i^t, j)$                   // acquire feature j
14        $total\_cost \leftarrow total\_cost + c_j$         // pay the cost of j
15        $r_{i,j}^t \leftarrow \frac{||Cert(\boldsymbol{x}_i^t) - Cert(\boldsymbol{x}_i^{t+1})||}{c_j}$
16        push $(\boldsymbol{x}_i^t, \boldsymbol{x}_i^{t+1}, j, r_{i,j}^t, \tilde{y}_i)$ into the replay memory
17        $t \leftarrow t + 1$
18        **if** *total_cost* $\geq B$ *or stop_condition() or no unknown feature* **then**
19           $terminate\_flag \leftarrow True$    // terminate if all the budget is used
20        **if** *update_condition()* **then**
21           train_batch $\leftarrow$ random mini-batch from the replay memory
22           update P, Q, and target Q networks using train_batch   // Jointly train P & Q
23     **end**
24 **end**

---

Table 1: The summary of datasets and experimental settings.

| Dataset | Instances | Features | Classes | P-Net Architecture | Q-Net Architecture |
|---|---|---|---|---|---|
| **MNIST** (LeCun et al., 1998) | 70000 | 784 | 10 | $[512, 512, 128, 64]$ | $[512 + 512, 512 + 256, 128 + 65, 64 + 16]$ |
| **LTRC** (Chapelle & Chang, 2011) | 34815 | 519 | 5 | $[128, 32]$ | $[128 + 128, 32 + 8]$ |
| **Diabetes** (Sec. 4.1) | 92062 | 45 | 3 | $[64, 32, 16]$ | $[64 + 64, 32 + 16, 16 + 10]$ |

## 4   RESULTS AND EXPERIMENTS

### 4.1   DATASETS AND EXPERIMENTS

We evaluated the proposed method on three different datasets: MNIST handwritten digits (LeCun et al., 1998), Yahoo Learning to Rank (LTRC) (Chapelle & Chang, 2011), and a health informatics dataset. The MNIST dataset is used as it is a widely used benchmark. For this dataset, we assume equal feature acquisition cost of 1 for all features. It is worth noting that we are considering the permutation invariant setup for MNIST where each pixel is a feature discarding the spatial information. Regarding the LTRC dataset, we use feature acquisition costs provided by Yahoo! that corresponding to the computational cost of each feature. Furthermore, we evaluated our method using a real-world health dataset for diabetes classification where feature acquisition costs and budgets are natural and essential to be considered. The national health and nutrition examination survey (NAHNES) data (nha, 2018) was used for this purpose. A feature set including: ($i$) demographic information (age, gender, ethnicity, etc.), ($ii$) lab results (total cholesterol, triglyceride, etc.), ($iii$) examination data (weight, height, etc.) , and ($iv$) questionnaire answers (smoking, alcohol, sleep habits, etc.) is used here. An expert with experience in medical studies is asked to suggest costs for each feature based on the overall financial burden, patient privacy, and patient inconvenience. Finally, the fasting glucose values were used to define three classes: normal, pre-diabetes, and diabetes based on standard threshold values. The final dataset consists of 92062 samples of 45 features.

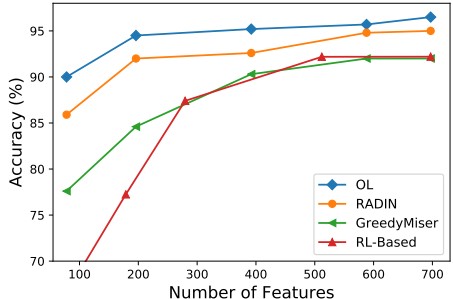
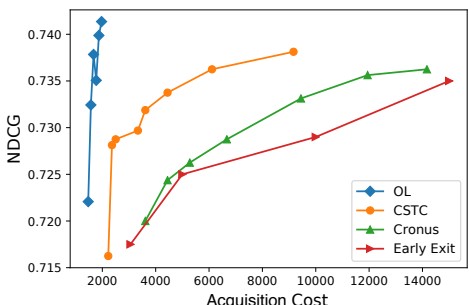

Figure 2: Evaluation of the proposed method on MNIST dataset. Accuracy vs. number of acquired features for OL, RADIN (Contardo et al., 2016), GreedyMiser (Xu et al., 2012), and a recent work based on reinforcement learning (RL-Based) (Janisch et al., 2017).

Figure 3: Evaluation of the proposed method on LTRC dataset. NDCG vs. cost of acquired features for OL, CSTC (Xu et al., 2014), Cronus (Chen et al., 2012), and Early Exit (Cambazoglu et al., 2010) approaches.

In the current study, we use reinforcement learning as an optimization algorithm, while processing data in a sequential manner. Throughout all experiments, we used fully-connected neural networks with ReLU non-linearity and dropout applied to hidden layers. We apply MC dropout sampling using 1000 evaluations of the predictor network for confidence measurements and finding the reward values. Meanwhile, 100 evaluations are used for prediction at test-time. We selected these value for our experiments as it showed stable prediction and uncertainty estimates. Each dataset was randomly splitted to 15% for test, 15% for validation, and the rest for train. During the training and validation phase, we use the random exploration mechanism. For comparison of the results with other work in the literature, as they are all offline methods, the random exploration is not used during the feature acquisition. However, intuitively we believe in datasets with non-stationary distributions, it may be helpful to use random exploration as it helps to capture concept drift. Furthermore, we do model training multiple time for each experiment and average the outcomes. It is also worth noting that, as the proposed method is incremental, we continued feature acquisition until all features were acquired and reported the average accuracy corresponding to each feature acquisition budget.

Table 1 presents a summary of datasets and network architectures used throughout the experiments. In this table, we report the number of hidden neurons at each network layer of the P and Q networks. For the Q-Network architecture, the number of neurons in each hidden layer is reported as the number of shared neurons from the P-Network plus the number of neurons specific to the Q-Network.

## 4.2 PERFORMANCE OF THE PROPOSED APPROACH

Figure 2 presents the accuracy versus acquisition cost curve for the MNIST dataset. Here, we compared results of the proposed method (OL) with a feature acquisition method based on recurrent neural networks (RADIN) (Contardo et al., 2016), a tree-based feature acquisition method (GreedyMiser) (Xu et al., 2012), and a recent work using reinforcement learning ideas (RL-Based) (Janisch et al., 2017). As it can be seen from this figure, our cost-sensitive feature acquisition method achieves higher accuracies at a lower cost compared to other competitors. Regarding the RL-Based method, (Janisch et al., 2017), to make a fair comparison, we used the similar network sizes and learning algorithms as with the OL method. Also, it is worth mentioning that the RL-based curve is the result of training many models with different cost-accuracy trade-off hyper-parameter values, while training the OL model gives us a complete curve. Accordingly, evaluating the method of (Janisch et al., 2017) took more than 10 times compared to OL.

Figure 3 presents the accuracy versus acquisition cost curve for the LTRC dataset. As LTRC is a ranking dataset, in order to have a fair comparison with other work in the literature, we have used the normalized discounted cumulative gain (NDCG) (Järvelin & Kekäläinen, 2002) performance measure. In short, NDCG is the ratio of the discounted relevance achieved using a suggested ranking method to the discounted relevance achieved using the ideal ranking. Inferring from Figure 3, the

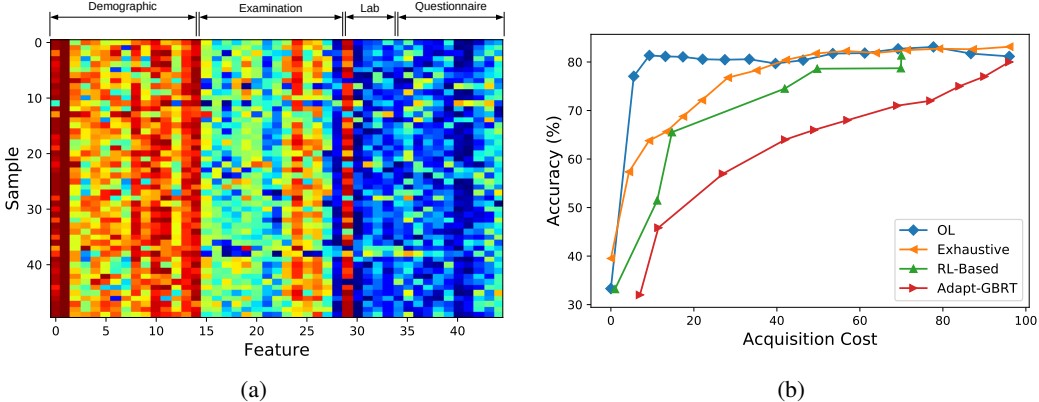

(a)                                                                 (b)

Figure 4: Evaluation of the proposed method on the diabetes dataset. (a) Visualization of feature acquisition orders for 50 test samples (warmer colors represent more priority). (b) Accuracy vs. cost of acquired features for this paper (OL), an exhaustive sensitivity-based method (Exhaustive) (Early et al., 2016a), the method suggested by Janisch et al. (2017) (RL-Based), and using gating functions and adaptively trained random forests (Nan & Saligrama, 2017) (Adapt-GBRT).

proposed method is able to achieve higher NDCG values using a much lower acquisition budget compared to tree-based approaches in the literature including CSTC (Xu et al., 2014), Cronus (Chen et al., 2012), and Early Exit (Cambazoglu et al., 2010).

Figure 4a shows a visualization of the OL feature acquisition on the diabetes dataset. In this figure, the y-axis corresponds to 50 random test samples and the x-axis corresponds to each feature. Here, warmer colors represent features that were acquired with more priority and colder colors represent less acquisition priority. It can be observed from this figure that OL acquires features based on the available context rather than having a static feature importance and ordering. It can also be seen that OL gives more priority to less costly and yet informative features such as demographics and examinations. Furthermore, Figure 4b demonstrates the accuracy versus acquisition cost for the diabetes classification. As it can be observed from this figure, OL achieves a superior accuracy with a lower acquisition cost compared to other approaches. Here, we used the exhaustive feature query method as suggested by Early et al. (2016a) using sensitivity as the utility function, the method suggested by Janisch et al. (2017) (RL-Based), as well a recent paper using gating functions and adaptively trained random forests (Nan & Saligrama, 2017) (Adapt-GBRT).

## 4.3 ANALYSIS

### 4.3.1 ABLATION STUDY

In this section we show the effectiveness of three ideas suggested by this paper i.e, using model uncertainty as a feature-value measure, representation sharing between the P and Q networks, and using MC-dropout as a measure of prediction uncertainty. Additionally, we study the influence of the available budget on the performance of the algorithm. In these experiments, we used the diabetes dataset. A comparison between the suggested feature-value function (OL) in this paper with a traditional feature-value function (RL-Based) was presented in Figure 2 and Figure 4b. We implemented the RL-Based method such that it is using a similar architecture and learning algorithm as the OL, while the reward function is simply the the negative of feature costs for acquiring each feature and a positive value for making correct predictions. As it can be seen from the comparison of these approaches, the reward function suggested in this paper results in a more efficient feature acquisition.

In order to demonstrate the importance of MC-dropout, we measured the average of accuracy at each certainty value. Statistically, confidence values indicate the average accuracy of predictions (Guo et al., 2017). For instance, if we measure the certainty of prediction for a group of samples to be 90%, we expect to correctly classify samples of that group 90% of the time. Figure 5 shows the average prediction accuracy versus the certainty of samples reported using the MC-dropout method (using 1000 samples) and directly using the softmax output values. As it can be inferred from this figure, MC-dropout estimates are highly accurate, while softmax estimates are mostly over-confident

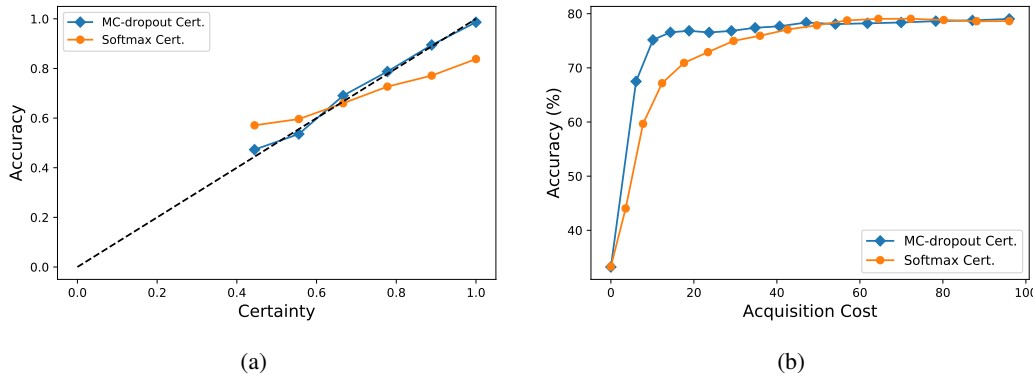

(a)                                    (b)

Figure 5: (a) The average prediction accuracy versus the certainty of samples reported using the MC-dropout method (1000 samples) and directly using the softmax output values. (b) The accuracy versus cost curves for using the MC-dropout method and directly using the softmax output values.

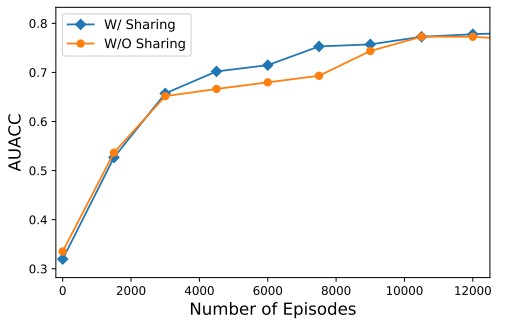

Figure 6: The speed of convergence using the suggested sharing between the P and Q networks (W/ Sharing) compared with not using the sharing architecture (W/O Sharing).

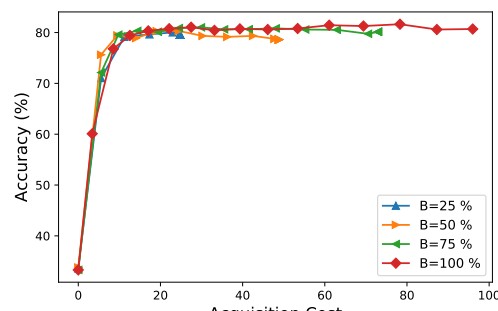

Figure 7: Accuracy versus cost curves achieved using different budget levels: 25 %, 50%, 75%, and 100% of the cost of acquiring all features.

and inaccurate. Note that the accuracy of certainty estimates are crucially important to us as any inaccuracy in these values results in having inaccurate reward values. Figure 5b shows the accuracy versus cost curves that the suggested architecture achieves using the accurate MC-dropout certainty estimates and using the inaccurate softmax estimates. It can be seen from this figure that more accurate MC-dropout estimates are essential.

Figure 6 demonstrates the speed of convergence using the suggested sharing between the P and Q networks (W/ Sharing) as well as not using the sharing architecture (W/O Sharing). Here, we use the normalized area under the accuracy-cost curve (AUACC) as measure of acquisition performance at each episode. Please note that we adjust the number of hidden neurons such that the number of Q-Network parameters is the same for each corresponding layer between the two cases. As it can be seen from this figure, the suggested representation sharing between the P and Q networks increases the speed of convergence.

Figure 7 shows the performance of the OL method having various limited budgets during the operation. Here, we report the accuracy-cost curves for $25\%$, $50\%$, $75\%$, and $100\%$ of the budget required to acquire all features. As it can be inferred from this figure, the suggested method is able to efficiently operate at different enforced budget constraints.

### 4.3.2   CONVERGENCE ANALYSIS

Figure 8a and 8b demonstrate the validation accuracy and AUACC values measured during the processing of the data stream at each episode for the MNIST and Diabetes datasets, respectively.

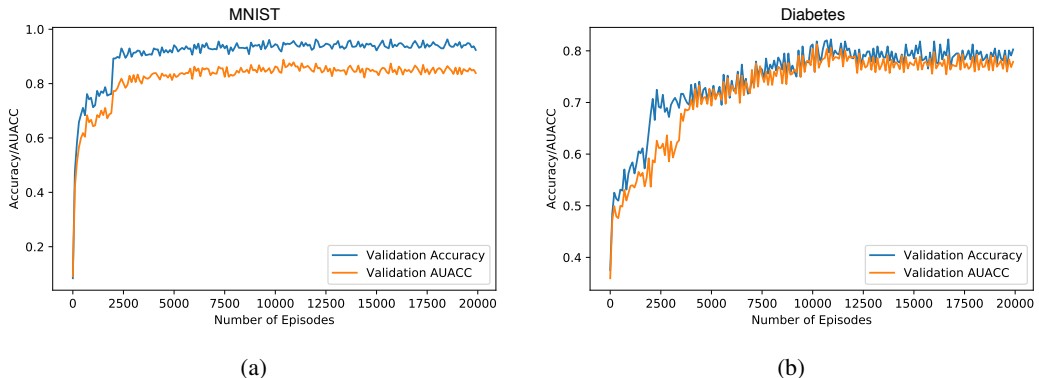

Figure 8: The validation set accuracy and AUACC values versus the number of episodes for the (a) MNIST and (b) Diabetes datasets.

As it can be seen from this figure, as the algorithm observes more data samples, it achieves higher validation accuracy/AUACC values, and it eventually converges after a certain number of episodes. It should be noted that, in general, convergence in reinforcement learning setups is dependent on the training algorithm and parameters used. For instance, the random exploration strategy, the update condition, and the update strategy for the target Q network would influence the overall time behavior of the algorithm. In this paper, we use conservative and reasonable strategies as reported in Section 3.2 that results in stable results across a wide range of experiments.

## 5 CONCLUSION

In this paper, we proposed an approach for cost-sensitive learning in stream-based settings. We demonstrated that certainty estimation in neural network classifiers can be used as a viable measure for the value of features. Specifically, variations of the model certainty per unit of the cost is used as measure of feature value. In this paradigm, a reinforcement learning solution is suggested which is efficient to train using a shared representation. The introduced method is evaluated on three different real-world datasets representing different applications: MNIST digits recognition, Yahoo LTRC web ranking dataset, and diabetes prediction using health records. Based on the results, the suggested method is able to learn from data streams, make accurate predictions, and effectively reduce the prediction-time feature acquisition cost.

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
