# OpenReview forum: "Opportunistic Learning: Budgeted Cost-Sensitive Learning from Data Streams"
_ICLR.cc/2019/Conference_

### Official Review · AnonReviewer2 · 2018-10-27
**budgeted feature acquisition to train networks - seems similar to RADIN**

**Rating:** 7
**Confidence:** 4

**Review:**

This paper presents a novel method for budgeted cost sensitive learning from Data Streams.
This paper seems very similar to the work of Contrado’s RADIN algorithm which similarly evaluates sequential datapoints with a recurrent neural network by adaptively “purchasing” the most valuable features for the current datapoint under evaluation according to a budget.

In this process, a sample (S_i) with up to “d” features arrives for evaluation.  A partially revealed feature vector x_i arrives at time “t” for consideration.  There seems to exist a set of “known features” that that are revealed “for free” before the budget is considered (Algorithm 1).  Then while either the budget is not exhausted or some other stopping condition is met features are sequentially revealed either randomly (an explore option with a decaying rate of probability) or according to their cost sensitive utility.  When the stopping condition is reached, a prediction is made.  After a prediction is made, a random mini-batch of the partially revealed features is pushed into replay memory along with the correct class label and the P. Q, and target Q networks are updated.

The ideas of using a sequentially revealed vector of features and sequentially training a network are in Contrado’s RADIN paper.   The main novelty of the paper seems to be the use of MC dropout as an estimate of certainty in place of the softmax output layer and the methods of updating the P and Q networks.
The value of this paper is in the idea that we can learn online and in a cost sensitive way.  The most compelling example of this is the idea that a patient shows up at time “t” and we would like to make a prediction of disease in a cost sensitive way.  To this end I would have liked to have seen a chart on how well this algorithm performs across time/history.  How well does the algorithm perform on the first 100 patients vs the last 91,962-91,062 patients at what point would it make sense to start to use the algorithm (how much history is needed).

Am I correct in assuming there are some base features that are revealed “for free” for all samples?  If so how are these chosen?  If so how does the number of these impact the results?

In Contrado’s RADIN paper the authors explore both the MNIST dataset and others, including a medical dataset “cardio.”  Why did you only use RADIN as a comparison for the MNIST dataset and not the LTRC or diabetes dataset?  Did you actually re-implement RADIN or just take the numbers from their paper?  In which case, are you certain which MNIST set was used in this paper? (it was not as well specified as in your paper).

With respect to the real world validity of the paper, given that the primary value of the paper has to do with cost sensitive online learning, it would have been better to talk more about the various cost structure and how those impact the value of your algorithm.  For the first example, MNIST, the assumed uniform cost structure is a toy example that equates feature acquisition with cost.  The second example uses computational cost vs relevance gain.  This would just me a measure of computational efficiency, in which case all of the computational cost of running the updates to your networks should also be considered as cost.  With respect to the third proprietary diabetes dataset, the costs are real and relevant, however there discussion of these are given except to say that you had a single person familiar with medical billing create them for you (also the web address you cite is a general address and does not go to the dataset you are using).

 In reality, these costs would be bundled.  You say you estimate the cost in terms of overall financial burden, patient privacy and patient inconvenience.  Usually if you ask the patient to fill out a survey it has multiple questions, so for the same cost you get all the answers.  Similarly if you do a blood draw and test for multiple factors the cost to the patient and the hospital are paid for the most part upfront.  It is not realistic to say that the cost of asking a patient a questions is 1/20th of the cost of the survey.  The first survey question asked would be more likely 90-95% of the cost with each additional question some incremental percentage.  To show the value of your work, a better discussion of the cost savings would be appreciated.

---

> ### Author Response · Authors · 2018-11-19
> **To Reviewer 3 (2/2)**
>
>
> *Comment: “In Contrado’s RADIN paper the authors explore both the MNIST dataset and others, including a medical dataset “cardio.” Why did you only use RADIN as a comparison for the MNIST dataset and not the LTRC or diabetes dataset? Did you actually re-implement RADIN or just take the numbers from their paper? In which case, are you certain which MNIST set was used in this paper? (it was not as well specified as in your paper).”
>
> We have compared our results with Contrado’s results as reported on the RADIN paper. The reason behind this was the fact that RADIN is consisting of many components and parameters which makes reproducing their results for our comparisons with RADIN very difficult. We would be glad to include comparisons with RADIN on other datasets, if the reviewer could point us to an open source implementation of RADIN.
> Regarding the reviewer’s comment on which samples of the MNIST was used for training/validation/test: we use the standard MNIST separation using the provided train set for our train and validation, and the MNIST test set is used for testing the suggested algorithm.
>
> -----------------------------------------------------------------------------------------------
> *Comment: “With respect to the real world validity of the paper, given that the primary value of the paper has to do with cost sensitive online learning, it would have been better to talk more about the various cost structure and how those impact the value of your algorithm...”
>
> We agree with the reviewer that it is very important to consider cost structures in real-world scenarios. However, a deep study of any specific cost structure (e.g., in specific healthcare problems) is itself an area of research and any problem would require an in-depth study. In this paper, we introduced a general formulation for the problem for cost-sensitive feature acquisition from stream data that is evaluated on different applications. However, a deeper study of any specific cost structure would require integrating domain expertise and is out of the scope of this study.
>
> -----------------------------------------------------------------------------------------------
> *Comment: “the web address you cite is a general address and does not go to the dataset you are using”
>
> The web address provided contains links to the dataset download page (the “Questionnaires, Datasets, and Related Documentation” option on the left sidebar). Additionally, we plan to publish the dataset preprocessing source code to help other future work to reproduce and compare our results.
>
>
> -----------------------------------------------------------------------------------------------
> *Comment: “In reality, these costs would be bundled...To show the value of your work, a better discussion of the cost savings would be appreciated.”
>
> The current formulation presented in this paper allows for having bundled feature sets. In this case, each action would be acquiring a bundle and the reward function is evaluated for the acquisition of this bundle by measuring the variations of uncertainty before and after acquiring the bundle. As suggested by the reviewer, we have added a discussion on this to the revised paper:
> “In our experiments, for the simplicity of presentation, we assume that all features are independently acquirable at a certain cost, while in many scenarios, features are bundled and acquired together (e.g., certain clinical measurements). However, it should be noted that the current formulation presented in this paper allows for having bundled feature sets. In this case, each action would be acquiring each bundle and the reward function is evaluated for the acquisition of the bundle by measuring the variations of uncertainty before and after acquiring the bundle.”

---

> ### Author Response · Authors · 2018-11-19
> **To Reviewer 3 (1/2)**
>
>
> Thank you for reviewing the manuscript and helpful comments. Please find a point-to-point response to your comments in the following.
>
> -----------------------------------------------------------------------------------------------
> * Comment: “The ideas of using a sequentially revealed vector of features and sequentially training a network are in Contrado’s RADIN paper.”
>
> We agree with the reviewer that having sequentially revealed vectors is in common between the earlier work by Contrado (RADIN) and the current study (OL). However, we believe that RADIN and OL are significantly different from each other in the idea, architecture, and implementation. Specifically:
>
> - RADIN approaches the problem by looking into the feature acquisition process as a time sequence of acquisitions. However, the suggested method is modeling the utility of actions given the current state regardless of the previous actions. From this perspective, RADIN can be considered a time-series approach, while OL is a reinforcement learning approach using a time-invariant policy.
>
> - RADIN defines a cost function consisting of two terms weighted by a hyper-parameter: a classification loss and a feature acquisition cost. However, the introduced method in this paper is using the variations of model uncertainty as a value function of eq (7) being used in making decisions.
>
> - RADIN is using a recurrent neural network (RNN) architecture, while OL is based on reinforcement learning and deep Q learning algorithms.
>
> - The suggested method is designed to operate as an online learning algorithm, while RADIN is not studying this case.
>
> -----------------------------------------------------------------------------------------------
> * Comment: “I would have liked to have seen a chart on how well this algorithm performs across time/history. How well does the algorithm perform on the first 100 patients vs the last 91,962-91,062 patients at what point would it make sense to start to use the algorithm (how much history is needed).”
>
> Thank you for suggesting this. We have included a new section to discuss this (see Section 4.3.2 and Fig. 8ab).
>
> We have also added the following explanation in the results section (see Section 4.3.2):
> “Figure 8a and 8b demonstrate the validation accuracy and AUACC values measured during the processing of the data stream at each episode for the MNIST and Diabetes datasets, respectively. As it can be seen from this figure, as the algorithm observes more data samples, it achieves higher validation accuracy/AUACC values, and it eventually converges after a certain number of episodes. It should be noted that, in general, convergence in reinforcement learning setups is dependent on the training algorithm and parameters used. For instance, the random exploration strategy, the update condition, and the update strategy for the target Q network would influence the overall time behavior of the algorithm. In this paper, we use conservative and reasonable strategies as reported in Section 3.2 that results in stable results across a wide range of experiments.”
>
> -----------------------------------------------------------------------------------------------
> *Comment: “Am I correct in assuming there are some base features that are revealed “for free” for all samples? If so how are these chosen? If so how does the number of these impact the results?”
>
> In our experiments, we are not assuming any feature will be available for free. However, the formulation presented in this paper accommodates the case where features are available for free. In order to clarify this issue and prevent any confusion to our readers, we added the following explanation to the paper:
>
> “In this algorithm, if any features are available for free we include them in the initial feature vector; otherwise, we start with all features not being available initially.”
> Also, the algorithm box is revised by adding a comment to Line 4:
> “$x_i^t$ <- known features of  S_i // if there are any features available”

---

### Official Review · AnonReviewer3 · 2018-11-05
**Interesting approach with a confused exposition**

**Rating:** 6
**Confidence:** 4

**Review:**

I like the approach, however: I consider the paper to be poorly written.  The presentation needs to be improved for me to find it acceptable.

It presents a stream-oriented (aka online) version of the algorithm, but experiments treat the algorithm as an offline training algorithm.  This is particularly critical in this area because feature acquisition costs during the "warm-up" phase are actual costs, and given the inherent sample complexity challenges of reinforcement learning, I would expect them to be significant in practice.  This would be fine if the setup is "we have a fixed offline set of examples where all features have been acquired (full cost paid) from which we will learn a selector+predictor for test time".

The algorithm 1 float greatly helped intelligibility, but I'm left confused.
  * Is this underlying predictor trained simultaneously to the selector?
        * Exposition suggests yes ("At the same time, learning should take place by updating the model while maintaining the budgets."), but algorithm block doesn't make it obvious.
        * Maybe line 21 reference to "train data" refers to the underlying predictor.
  * Line 16 pushes a value estimate into the replay buffer based upon the current underlying predictor, but:
        * this value will be stale when we dequeue from the replay buffer if the underlying predictor has changed, and
        * we have enough information stored in the replay buffer to recompute the value estimate using the new predictor, but
        * this is not discussed at all.

Also, I'm wondering about the annealing schedule for the exploration parameter (this is related to my concern that the
algorithm is not really an online algorithm).  The experiments are all silent on the "exploration" feature acquisition cost.  Furthermore I'm wondering: when you do the test evaluations, do you set exploration to 0?

I also found the following disturbing: "It is also worth noting that, as the proposed method is
incremental, we continued feature acquisition until all features were acquired and reported the average
accuracy corresponding to each feature acquisition budget."  Does this mean the underlying predictor was trained on data
that it would not have if the budget constraint were strictly enforced?

---

> ### Author Response · Authors · 2018-11-19
> **To Reviewer 2 (2/2)**
>
>
> *Comment: Also, I'm wondering about the annealing schedule for the exploration parameter (this is related to my concern that the algorithm is not really an online algorithm). The experiments are all silent on the "exploration" feature acquisition cost. Furthermore I'm wondering: when you do the test evaluations, do you set exploration to 0?
>
> In an online learning setup, data becomes available sequentially and the goal for an online learner is to update its hypothesis as more data is being observed. There are two main considerations for an online method. First, data is not provided or can be stored as a batch. Second, the hypothesis should be refined incrementally as more observations take place.
>
> Regarding the reviewer’s concern about annealing, annealing is a standard approach widely used in the literature helping early steps of optimization. We believe that the suggested algorithm is online because, initially, there is no viable alternative strategy to follow due to the limited number of samples. However, as we observe more samples, we anneal the random decisions and try to use the captured knowledge instead. In this respect, the suggested algorithm is online according to the definition above.
>
> Regarding the exploration probability used in our experiments: during the training and validation phase, we use the random exploration mechanism. However, for the comparison of the results with other work in the literature, as they are all offline methods, we decided to not to do the exploration.
>
> In order to address the reviewer’s comment, we added the following explanation to the revised paper:
> “During the training and validation phase, we use the random exploration mechanism. However, for the comparison of the results with other work in the literature, as they are all offline methods, the random exploration is not used during the feature acquisition.”
>
> -----------------------------------------------------------------------------------------------
> *Comment: I also found the following disturbing: "It is also worth noting that, as the proposed method is incremental, we continued feature acquisition until all features were acquired and reported the average accuracy corresponding to each feature acquisition budget." Does this mean the underlying predictor was trained on data that it would not have if the budget constraint were strictly enforced?
>
> In order to address the reviewer’s concern, we conducted experiments using different enforced budgets (see Fig. 7). In summary, according to our experiments, the suggested method is able to efficiently operate at different enforced budget constraints.
>
> We have also included the following discussion to the paper:
> “Figure 7 shows the performance of the OL method having various limited budgets during the operation. Here, we report the accuracy-cost curves for 25%, 50%, 75%, and 100% of the budget required to acquire all features. As it can be inferred from this figure, the suggested method is able to efficiently operate at different enforced budget constraints.”

---

> > ### Comment · AnonReviewer3 · 2018-11-25
> > **We're talking past each other a bit here, but that's ok**
> >
> > >> Regarding the reviewer’s concern about annealing, annealing is a standard approach widely used in the literature helping early steps of optimization.
> >
> > My concern is not about annealing per se.  Rather, the literature of online learning focuses on cumulative regret, which the manuscript does not address.  ("Optimization" != "online learning").  I suspect you mean "online" as in "receiving examples one-at-a-time", but this distinction should be clearer.
> >
> > >> Regarding the exploration probability used in our experiments: during the training and validation phase, we use the random exploration mechanism. However, for the comparison of the results with other work in the literature, as they are all offline methods, we decided to not to do the exploration.
> >
> > The Achilles' Heel of reinforcement learning is sample complexity, and our (lack of good) exploration algorithms is central to the problem.  By turning off the exploration for test you have elided this difficulty ("in real life, when are you testing?").   It's not a fatal flaw, because this paper uses the "reinforcement learning as an optimization algorithm" and "online learning as sequential data presentation" perspectives, but clarifying this would improve the exposition.
> >
> > >> In order to address the reviewer’s concern, we conducted experiments using different enforced budgets (see Fig. 7). In summary, according to our experiments, the suggested method is able to efficiently operate at different enforced budget constraints.
> >
> > This is great.  What this tells me is that in the experiments most of the features are not useful, so placing a hard upper limit on feature acquisition during learning is not damaging.

---

> > > ### Author Response · Authors · 2018-11-25
> > > **Re: We're talking past each other a bit here, but that's ok**
> > >
> > >
> > > * Comment: "My concern is not about annealing per se.  Rather, the literature of online learning focuses on cumulative regret, which the manuscript does not address.  ("Optimization" != "online learning").  I suspect you mean "online" as in "receiving examples one-at-a-time", but this distinction should be clearer."
> > >
> > > As requested by the reviewer, we have added the following clarification to the revised version (the first paragraph of the Introduction):
> > > "Here, by online we mean processing samples one at a time as they are being received."
> > >
> > > --------------------------------------
> > > * Comment: "The Achilles' Heel of reinforcement learning is sample complexity, and our (lack of good) exploration algorithms is central to the problem.  By turning off the exploration for test you have elided this difficulty ("in real life, when are you testing?").   It's not a fatal flaw, because this paper uses the "reinforcement learning as an optimization algorithm" and "online learning as sequential data presentation" perspectives, but clarifying this would improve the exposition."
> > >
> > > Thank you for pointing this out. We have included a clarification in the revised paper (see the second paragraph of Sec 4):
> > > "In the current study, we use reinforcement learning as an optimization algorithm, while processing data in a sequential manner."
> > >
> > > We have also added a brief discussion of using random exploration during the prediction (the second paragraph of Sec 4):
> > > "However, intuitively we believe in datasets with non-stationary distributions, it may be helpful to use random exploration as it helps to capture concept drift."

---

> ### Author Response · Authors · 2018-11-19
> **To Reviewer 1 (1/2)**
>
>
> Thank you for reviewing the manuscript and helpful comments. Please find a point-to-point response to your comments in the following.
> -----------------------------------------------------------------------------------------------
> * Is this underlying predictor trained simultaneously to the selector?
> * Exposition suggests yes ("At the same time, learning should take place by updating the model while maintaining the budgets."), but algorithm block doesn't make it obvious.
>
> Yes, the predictor is trained jointly with the feature-value estimator. In algorithm block, Line 22 is related to updating P and Q networks at the same time on a training batch sampled from the relay memory. In order to clarify this, we have added the following comment to the algorithm box in the revised version:
> “update P, Q, and target Q networks using train-batch // Jointly train P & Q”
>
> -----------------------------------------------------------------------------------------------
> * Maybe line 21 reference to "train data" refers to the underlying predictor.
> * Line 16 pushes a value estimate into the replay buffer based upon the current underlying predictor, but:
>     * this value will be stale when we dequeue from the replay buffer if the underlying predictor has changed, and
>     * we have enough information stored in the replay buffer to recompute the value estimate using the new predictor, but
>     * this is not discussed at all.
>
> Here, train data refers to a batch of samples from the experience replay memory. Each item in the replay memory is a tuple of: a feature vector before and after the acquisition, the action taken, reward received for that action, and the ground-truth label corresponding to that feature vector (see Line 16 of the algorithm box).
>
> Regarding the reviewer’s concern about issues with having stale predicted values, in this paper, we prevent this by storing the ground-truth label in the replay buffer and by recomputing the predictions before updating the parameters. This way, we always use the most up-to-date results. However, as the ground truth label may not be available during the feature acquisition, in our final implementation, we use a temporary buffer to store experiences without the label and we push them along with the ground truth label as soon as the feature acquisition is finished and the ground-truth label is available. To simplify the presentation of the algorithm, we decided to omit the temporary buffering trick from the algorithm box and assumed that the labels are available. If the reviewer believes that including this in the algorithm would be helpful, we would be glad to include this.
>
> We have discussed this in the second-to-last paragraph of Section 3.1:
> “It is worth noting that, in Algorithm 1, to simplify the presentation, we assumed that ground-truth labels are available at the beginning of each episode. However, in the actual implementation, we store experiences within an episode in a temporary buffer, excluding the label. At last, after the termination of the feature acquisition procedure, a prediction is being made and upon the availability of label for that sample, the temporary experiences along with the ground-truth label are pushed to the experience replay memory.”

---

> > ### Comment · AnonReviewer3 · 2018-11-25
> > **Great response to these points**
> >
> > You have effectively incorporated the feedback from the points you quote in this response.

---

> > > ### Author Response · Authors · 2018-11-25
> > > **Re: Great response to these points**
> > >
> > > We are glad to hear that you found the revised version satisfactory!
> > > We would appreciate if you could reconsider the decision or let us know if you have any additional concern.

---

> > > > ### Comment · AnonReviewer3 · 2018-11-25
> > > > **I am upgrading my review**
> > > >
> > > > I had to write my response to your other response first.
> > > >
> > > > If it doesn't happen for you this time, I recommending focusing more on the train-time behaviour going forward.  I suspect you could discover an exploration algorithm that is empirically more effective than epsilon-greedy.  Furthermore, as a practitioner I care most about 1) cumulative regret [total acquisition cost including learning], 2) tracking non-stationary environments.  #2 includes the availability of new features as well as changes in cost or effectiveness of existing features.

---

> > > > > ### Author Response · Authors · 2018-11-25
> > > > > **Re: I am upgrading my review**
> > > > >
> > > > > We agree with the reviewer that having efficient exploration algorithms would be crucial in this setting. We believe a more extensive study of different exploration techniques would be a great subject for any future work.

---

### Official Review · AnonReviewer1 · 2018-11-05
**nice results but limited novelty**

**Rating:** 6
**Confidence:** 4

**Review:**

The paper presents a RL approach for sequential feature acquisition in a budgeted learning setting, where each feature comes at some cost and the goal is to find a good trade-off between accuracy and cost. Starting with zero feature, the model sequentially acquires new features to update its prediction and stops when the budget is exhausted. The feature selection policy is learned by deep Q-learning. The authors have shown improvements over several prior approaches in terms of accuracy-cost trade-off on three datasets, including a real-world health dataset with real feature costs.

While the results are nice, the novelty of this paper is limited. As mentioned in the paper, the RL framework for sequential feature acquisition has been explored multiple times. Compared to prior work, the main novelty in this paper is a reward function based on better calibrated classifier confidence. However, ablations study on the reward function is needed to understand to what extent is this helpful.

I find the model description confusing.
1. What is the loss function? In particular, how is the P-Network learned? It seems that the model is based on actor-critic algorithms, but this is not clear from the text.
2. What is the reward function? Only immediate reward is given.
3. What is the state representation? How do you represent features not acquired yet?

It is great that the authors have done extensive comparison with prior approaches; however, I find more ablation study needed to understand what made the model works better. There are at least 3 improvements: 1) using proper certainty estimation; 2) using immediate reward; 3) new policy architecture. Right now not clear which one gives the most improvement.

Overall, this paper has done some nice improvement over prior work along similar lines, but novelty is limited and more analysis of the model is needed.

---

> ### Author Response · Authors · 2018-11-19
> **To Reviewer 1**
>
>
> Thank you for reviewing the manuscript and helpful comments. Please find a point-to-point response to your comments in the following.
>
> --------------------------------------------
> *Comment: “1. What is the loss function? In particular, how is the P-Network learned? It seems that the model is based on actor-critic algorithms, but this is not clear from the text.”
>
> The loss function used for the P-Network is a cross-entropy loss as a typical loss used for classification tasks. For training the Q-Network we use mean squared error (MSE) between the estimated reward and the observed reward values. Please note that the replay buffer is used to sample batches of feature vectors, labels, actions, and reward values required to measure P- and Q-Network losses.
>
> We added the following clarification to the revised paper (see Sec. 3.2):
> “Cross-entropy and mean squared error (MSE) loss functions were used as the objective functions for the P and Q networks, respectively.”
>
> --------------------------------------------
> *Comment: “2. What is the reward function? Only immediate reward is given.”
>
> The reward function which is suggested by this paper is presented in Eq. (7). Here, we are using epsilon-greedy explorations and Bellman equations to fit the action-value function. It allows the general formulation of non-immediate and accumulated rewards through a discount factor.
> Intuitively, the suggested reward function in Eq. (7) measures the expected change of model hypothesis corresponding to each feature acquisition action.
>
> --------------------------------------------
> *Comment: “3. What is the state representation? How do you represent features not acquired yet?”
>
> In this paper, each state is the current feature vector containing values for features that are acquired at that state.
> From the first paragraph of Section 3.1:
> “At each point, the current state is defined as the current realization of the feature vector (i.e., $\bm{x}_i^t$) for a given instance.”
>
> We use NaN (not a number) values to internally represent the features that are not available. However, the implementation we use replaces the NaN values with zeros during the forward/backward computation. We believe it is an efficient approach compared to using separate mask vectors to represent missing features as it reduces the memory and I/O overheads.
>
> We included a brief explanation in the revised version (See Sec. 3.2):
> “Note that, in our implementation, for efficiency reasons, we use NaN (not a number) values to represent features that are not available and impute them with zeros during the forward/backward computation.”
>
> --------------------------------------------
> *Comment: “It is great that the authors have done extensive comparison with prior approaches; however, I find more ablation study needed to understand what made the model works better.”
>
>
> Thank you for suggesting this. In the revised version, we added a new subsection (Sec. 4.3.1) to the results section entitled “ablation study”. In summary, it presents an ablation study and comparisons of:
>
> - Using the MC-Dropout certainty versus the uncalibrated softmax estimates. We compared the accuracy of the estimated certainty values achieved as well as the overall impact on the feature acquisition performance (see Fig. 5a and Fig. 5b of the revised version). As it can be seen from these figures, the idea of using MC-Dropout certainty plays a crucial rule in the performance of the proposed method.
>
> - Demonstrating the effectiveness of the suggested representation sharing between the P and Q networks (see Fig. 6) demonstrating that the representation sharing would result in a faster convergence.
>
> - We added an analysis of the suggested method under different enforced budget constraints (see Fig. 7). According to results, the suggested method is able to efficiently operate at different enforced budget constraints.
>
> - Regarding other ablation analysis suggested by the reviewer, we have comparisons of the suggested approach (OL) and a basic reinforcement learning based method (RL-based) in the comparison results presented in Section 4.2. Due to space considerations, in the revised version, we discussed this case in the ablation study section without reiterating the plots and by referring to Fig. 2 and Fig. 4b (see Sec 4.3.1):
> “A comparison between the suggested feature-value function (OL) in this paper with a traditional feature-value function (RL-Based) was presented in Figure 2 and Figure 4b. Here, RL-Based method is using a similar architecture and learning algorithm as the OL, while the reward function is simply the negative of feature costs for acquiring each feature and a positive value for making correct predictions. As it can be seen from the comparison of these approaches, the reward function suggested in this paper results in a more efficient feature acquisition.“

---

> > ### Comment · AnonReviewer1 · 2018-11-26
> > **thanks for the revision**
> >
> > Thanks for the revision! I appreciate the authors' efforts addressing reviewers' comments and have updated my score.
> >
> > Ablation study Figure 6:
> > Why is the training curve shown here? I believe the main objective is accuracy-cost trade-off so I was expecting a figure like Figure 5(b). The convergence rates look pretty similar to me anyway (considering the variance).
> >
> > Ablation study Figure 7:
> > Which dataset is this on? It seems that on this dataset a small number of features yields accuracy close to the full-features setting. I wonder if a static feature selection method would do as well.
> >
> > Also, just noticed, why are the baselines different for different datasets? i.e Figure 2, 3, 4(b) have different baselines.

---

> > > ### Author Response · Authors · 2018-11-26
> > > **Re: thanks for the revision**
> > >
> > >
> > > * Comment: "Ablation study Figure 6: Why is the training curve shown here? I believe the main objective is accuracy-cost trade-off so I was expecting a figure like Figure 5(b). The convergence rates look pretty similar to me anyway (considering the variance)."
> > >
> > > In Figure. 6, we show the speed of convergence with and without using the suggested representation sharing method. As it can be seen from this figure, the representation sharing would help the faster convergence. Here, after the convergence, the accuracy-cost curves would be very similar.
> > >
> > > Regarding your concern about the statistical significance of the results, the curves presented in this figure are average of 8 different randomly initialized runs. We believe that the representation sharing idea would help the convergence speed and worthy to be used in implementations of our work.
> > >
> > > ------------------------------
> > > * Comment: "Ablation study Figure 7: Which dataset is this on? It seems that on this dataset a small number of features yields accuracy close to the full-features setting. I wonder if a static feature selection method would do as well."
> > >
> > > As noted in the first paragraph of the ablation study, Diabetes dataset was used here. The purpose of this figure is demonstrating the effect of different enforced budgets on the over performance. As it can be seen from this figure, having enforced budget does not have a considerable influence on the exploration and training of the P and Q networks.
> > >
> > > Regarding your concern about the easiness of the task, We compared our results with other baselines in Figure 4b. As it can be seen in this figure, many other works that are better than static methods are not able to converge as fast as OL.
> > >
> > > ---------------------------
> > > * Comment: "Also, just noticed, why are the baselines different for different datasets? i.e Figure 2, 3, 4(b) have different baselines."
> > >
> > > We tried our best to use all the baselines on all figures; however, there are technical difficulties such as:
> > > - Exhaustive: this method is computationally very expensive, we could only run it on our smallest dataset.
> > > - Adapt-GBRT: the loss function and the source code provided by the authors are mostly appropriate for regression tasks or classification tasks with label ordering.
> > > - RADIN: this method has many hyper-parameters and we found it difficult to reimplement. We decided to report the results as presented in the original paper.
> > > - Tree-based approaches, such as EarlyExit, Cronus,  GreedyMiser, are usually less powerful than the Adapt-GBRT, so we decided to omit these comparisons for the Diabetes dataset to enhance the readability.

---

### Author Response · Authors · 2018-11-19
**Summary of changes**


We thank the reviewers for the constructive comments and suggestions. We believe that the suggested revisions enhanced the scientific quality of the manuscript significantly.

The summary of revisions is as follows:

- We added clarifications to different parts of the paper including: loss functions, algorithm box comments, explanation of the algorithm, implementation details, and so forth.

- A new section is included in the revised version which is dedicated to ablation study of: (i) certainty measurement used in this paper, (ii) the suggested representation sharing method, (iii) the proposed reward function, and (iv) the performance of the proposed method under different enforced budgets (see Section 4.3.1).

- We presented and discussed plots showing the performance of the algorithm during the operation at each episode.

- We added discussions on how to handle bundled features, how to represent missing values, the impacts of the annealing strategy, etc.

---

### Meta-Review · Area_Chair1 · 2018-12-13

**Confidence:** 5
**Recommendation:** Accept (Poster)

**Metareview:**

This paper presents a reinforcement learning approach for online cost-aware feature acquisition. The utility of each feature is measured in terms of expected variations of the model uncertainty (using MC dropout sampling as an estimate of certainty) which is subsequently used as a reward function in the reinforcement learning formulation. The empirical evaluations show improvements over prior approaches in terms of accuracy-cost trade-off on three datasets. AC can confirm that all three reviewers have read the author responses and have significantly contributed to the revision of the manuscript.

Initially, R1 and R2 raised important concerns regarding low technical novelty. R1 requested an ablation study to understand which of the following components gives the most improvement: 1) using proper certainty estimation; 2) using immediate reward; 3) new policy architecture. Pleased to report that the authors addressed the ablation study in their rebuttal and confirmed that MC-dropout certainty plays a crucial rule in the performance of the proposed method. R1 subsequently increased the assigned score to 6. R2 raised concerns about related prior work Contardo et al 2016, which similarly evaluates the most informative features given budget constraints with a recurrent neural network approach. After a long discussion and a detailed rebuttal, R2 upgraded the rating from below the threshold to 7, albeit acknowledging an incremental technical contribution. R3 raised important concerns regarding presentation clarity that were subsequently addressed by the authors. In conclusion, all three reviewers were convinced by the authors rebuttal and have upgraded their initial rating, and AC recommends acceptance of this paper – congratulations to the authors!